# LLM-Augmented Soft-Label Distillation and Cluster-Guided Alignment for Language-Based Audio Retrieval

## Abstract

Language-based audio retrieval involves fetching audio recordings from a database that most closely align with a provided text query. In this paper, we study language-based audio retrieval with a dual encoder and show that (i) soft-label distillation from an ensemble of retrieval teachers, (ii) LLM-driven caption augmentation (back-translation and caption mix for mixed audio), and (iii) cluster-guided auxiliary classification jointly improve robustness to non-binary audio-text correspondences. On CLOTHO dataset, our best single model reaches mAP@16 46.6, and a weighted ensemble attains 48.8 on the development test split. While cluster guidance yields mixed gains across backbones, ablations indicate consistent improvements under high correspondence ambiguity.

## 1 INTRODUCTION

Language-based audio retrieval is a task that requires retrieving audio recordings from a database that best matches a given textual query. This task is critical for applications such as content-based multimedia search, audio annotation, and cross-modal understanding, where aligning audio and text modalities in a shared semantic space is essential. Unlike traditional audio classification or tagging, language-based audio retrieval demands models that capture nuanced semantic relationships between free-form text descriptions and complex audio signals, which may contain overlapping or ambiguous acoustic concepts. Our approach builds on a dual encoder architecture with advanced techniques, such as distillation loss, LLM-based data augmentation, and auxiliary classification. These methods aim to enhance the model's generalization, robustness, and ability to capture fine-grained audio-text relationships. We summarize our contributions as follows:

- Soft-label distillation that targets non-binary audio–caption correspondences.
- Reproducible LLM-based augmentation pipeline for mixed-audio captions.
- Cluster-guided auxiliary heads that align audio with text topics; thorough ablations on topic granularity and teacher softness.

The remainder of this paper is organized as follows. Section 2 describes the proposed system in detail. Section 3 outlines the datasets, models, and training protocols. Finally, Section 4 presents the experimental results and describes our systems.

## 2 METHOD

Our system leverages a dual encoder architecture, where audio and text inputs are processed by separate encoders and aligned in a joint embedding space. We enhance this framework with contrastive learning, distillation loss, an auxiliary classification task, and data augmentation, as detailed below. The overall structure is illustrated in Figure 1.

### 2.1 CONTRASTIVE LEARNING

We employed a contrastive learning framework as the foundational approach to align audio and text representations. Contrastive learning seeks to create a joint embedding space where corresponding

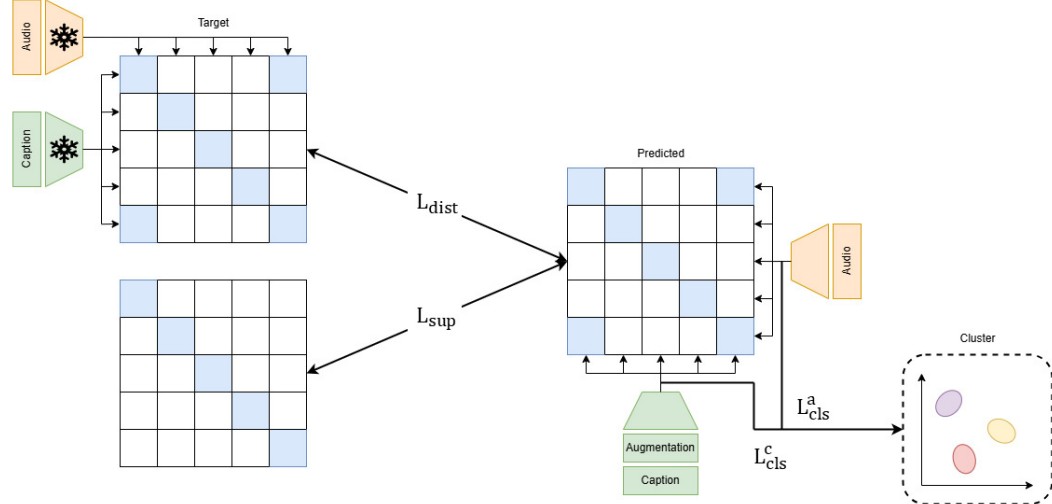

Figure 1: Overview of our system. The pretrained model is used to generate targets for the distillation loss. After the finetuning phase, clustering is performed separately on audio and text data to assign pseudo-labels. These pseudo-labels are used for an auxiliary classification task that guides re-finetuning.

audio-text pairs are closely aligned, while non-corresponding pairs are distanced (Koepke et al., 2022). This is accomplished by optimizing the InfoNCE loss, which maximizes the cosine similarity of matched audio-text embeddings and minimizes it for unmatched pairs within a batch. Let $\phi_a$ and $\phi_c$ denote the audio and text encoders, respectively, which map audio inputs $a_i$ and text captions $c_j$ to their respective embeddings. The similarity between an audio embedding $\phi_a(a_i)$ and a text embedding $\phi_c(c_j)$ is defined as the normalized cosine similarity:

$$C_{ij} = \frac{\phi_a(a_i)^T \cdot \phi_c(c_j)}{\|\phi_a(a_i)\|_2 \|\phi_c(c_j)\|_2},\qquad(1)$$

where $\| \cdot \|_2$ represents the L2 norm, ensuring unit-normalized embeddings. We compute softmax-normalized probabilities for audio-to-text and text-to-audio retrieval as:

$$q_a(a_i|c_j) = \frac{exp(C_{ij}/\tau)}{\sum_{k=1}^{N} exp(C_{kj}/\tau)},\qquad(2)$$

$$q_c(c_j|a_i) = \frac{exp(C_{ij}/\tau)}{\sum_{l=1}^{N} exp(C_{il}/\tau)},\qquad(3)$$

where $\tau > 0$ is a temperature parameter controlling the softness of the distribution. We used $\tau = 0.05$ in all our experiments. These probabilities reflect the model's confidence in matching audio $a_i$ to caption $c_j$ and vice versa, relative to other items in the batch. The supervised contrastive loss is the sum of cross-entropy losses between the predicted probabilities $(q_a, q_c)$ and the ground-truth distributions $(p_a, p_c)$, where $p_a$ and $p_c$ assign a probability of 1 to the positive pair and 0 to negative pairs:

$$L_{sup} = H(p_a, q_a) + H(p_c, q_c),\qquad(4)$$

where $H$ is the cross-entropy loss.

## 2.2 DISTILLATION LOSS

To address the binary correspondence assumption in audio retrieval datasets like CLOTHO, where captions may describe multiple recordings due to overlapping acoustic concepts or limited diversity, we adopted a distillation loss approach from the top-ranked DCASE 2024 Task 8 system (Primus

Table 1: System ID (SID) for various training configurations

| SID | Distill | Augmentation | Cluster label source |
|-----|---------|--------------|----------------------|
| 1 | X | X | X |
| 2 | O | X | X |
| 3 | O | O | X |
| 4 | O | O | Finetuned |
| 5 | O | O | BERTopic |

et al., 2024). This method uses soft correspondence probabilities from an ensemble of pretrained models to capture nuanced audio-text relationships, improving generalization. Formally, we first compute the similarity between audio embedding and text embedding as defined in Section 2.1. An ensemble of $M$ pretrained models generates soft correspondence probabilities by averaging their similarity scores:

$$\hat{C}_{ij} = \frac{1}{M} \sum_{m=1}^{M} C_{ij}^{m}. \tag{5}$$

These averaged similarities are used to compute soft probabilities in a knowledge distillation-like procedure:

$$\hat{p}_a(a_i|c_j) = \frac{exp(\hat{C}_{ij}/\tau)}{\sum_{k=1}^{N} exp(\hat{C}_{kj}/\tau)}, \tag{6}$$

$$\hat{p}_c(c_j|a_i) = \frac{exp(\hat{C}_{ij}/\tau)}{\sum_{l=1}^{N} exp(\hat{C}_{il}/\tau)}, \tag{7}$$

The distillation loss is calculated as the cross-entropy between these soft probabilities and the model's predicted probability:

$$L_{dist} = H(\hat{p}_a, q_a) + H(\hat{p}_c, q_c). \tag{8}$$

The total loss combines the supervised contrastive loss $L_{sup}$ with the distillation loss, weighted by $\lambda = 1.0$:

$$L = L_{sup} + \lambda L_{dist} \tag{9}$$

By leveraging these soft targets, the distillation loss enhances the model's ability to capture complex relationships between audio recordings and captions, improving its generalization across diverse audio-text pairs.

## 2.3 CLUSTER-BASED CLASSIFICATION

We propose a novel approach to enhance language-based audio retrieval by introducing an auxiliary classification task to further improve the model's representation learning. We perform clustering on all captions in the CLOTHO dataset to lay the foundation for an auxiliary task. We generate embedding for each caption and apply a clustering method similar to BERTopic (Grootendorst, 2022), which typically involves dimensionality reduction, such as UMAP (McInnes et al., 2018), followed by density-based clustering, such as HDBSCAN (McInnes et al., 2017), to group captions into semantically similar clusters. Each caption is thus assigned to a specific cluster, representing latent topics or semantic patterns within the captions.

To leverage the clustering results, we extend the model architecture by adding classification heads to both the text and audio encoders. The classification head for the text encoder is designed to predict the cluster label of the input caption, while the audio encoder's classification head predicts the cluster label of the corresponding caption. Specifically, the output of each encoder is processed through two sequential linear layers with a ReLU activation function between them, projecting the output to a vector with dimensions equal to the number of clusters. The intermediate linear layer has a dimension three times that of the input to enhance representation capacity. This setup encourages the audio encoder to learn representations that are aligned with the semantic clusters of the captions,

| SID | Audio model | Multiple annotation | | | Single annotation | | |
|---|---|---|---|---|---|---|---|
| | | mAP@10 | mAP@16 | mAP@10 | R@1 | R@5 | R@10 |
| 1 | PaSST | 39.45 | 42.08 | 35.47 | 23.35 | 52.5 | 65.07 |
| | EAT | 38.11 | 40.41 | 35.13 | 23.44 | 51.12 | 63.87 |
| | BEATs | 35.66 | 38.12 | 34.15 | 22.74 | 49.51 | 63.75 |
| 2 | PaSST | 43.75 | **46.62** | 39.32 | 26.81 | 56.61 | 70.07 |
| | EAT | 42.83 | 45.35 | 39.50 | 26.79 | 56.40 | 69.44 |
| | BEATs | 41.36 | 43.89 | 37.92 | 25.26 | 54.81 | 69.00 |
| 3 | PaSST | 43.56 | 46.41 | 39.92 | 27.20 | **57.84** | 70.74 |
| | EAT | 43.37 | 46.05 | **40.28** | **27.52** | 57.63 | **71.35** |
| | BEATs | 42.09 | 44.66 | 38.42 | 25.51 | 56.02 | 69.44 |
| 4 | PaSST | 43.61 | 46.39 | 39.92 | 27.2 | 57.21 | 70.24 |
| | EAT | 42.83 | 45.34 | 40.02 | 27.43 | 56.59 | 70.62 |
| | BEATs | 42.01 | 44.58 | 38.61 | 25.88 | 55.94 | 69.46 |
| 5 | PaSST | **43.79** | 46.50 | 39.58 | 26.66 | 57.38 | 70.14 |
| | EAT | 42.65 | 45.34 | 39.73 | 26.67 | 57.28 | 70.18 |
| | BEATs | 41.32 | 43.88 | 38.23 | 25.26 | 56.06 | 69.86 |
| Ensemble | | | | | | | |
| E1 | | **46.07** | **48.83** | 41.60 | 28.33 | 59.71 | 72.06 |
| E2 | | 46.05 | 48.78 | 41.58 | 28.34 | 59.87 | 72.23 |
| E3 | | 46.03 | 48.80 | 41.70 | **28.46** | 59.85 | 72.38 |
| E4 | | 46.04 | 48.79 | **41.72** | 28.38 | **60.02** | **72.46** |

Table 2: Retrieval performance of the models (first section) and the ensembled systems (second section). Note that SID stands for System ID, which is detailed in Table 1.

thereby enhancing the fine-grained alignment between audio and text. The total loss combines the supervised contrastive loss $L_{sup}$ from Section 2.1, the distillation loss $L_{dist}$ from Section 2.2, and the classification losses for the audio and text encoders, denoted $L_{cls}^a$ and $L_{cls}^c$, respectively:

$$L = L_{sup} + \lambda_1 L_{dist} + \lambda_2 (L_{cls}^a + L_{cls}^c) \tag{10}$$

In all experiments, we fixed $\lambda_1 = 1.0$ and $\lambda_2 = 0.05$ to balance the contributions of each loss term.

## 2.4 DATA AUGMENTATION

To enhance the diversity of captions for our text-grounded audio retrieval, we employed caption augmentation leveraging the capabilities of a large language model (LLM), specifically GPT-4o (Hurst et al., 2024). One of the key techniques utilized was **back-translation** (Sennrich et al., 2015). This method involves translating the original English captions into a randomly selected language and then translating them back into English. By doing so, back-translation generates captions that retain the same semantic meaning as the originals but feature varied linguistic expressions. In addition to back-translation, we implemented another augmentation technique called **LLM mix** (Wu et al., 2024) to further enrich our dataset. For this method, we randomly selected two audio-text pairs and combined their audio signals to create a new mixed audio sample. To generate a corresponding caption for this mixed audio, we utilized GPT-4o to intelligently merge the captions of the original audio-text pairs. With LLM mix, we created 50,000 new audio-text pairs, adding substantial variety to our dataset

## 3 EXPERIMENTS

The following subsections provide comprehensive details on the datasets, models, and training protocols to ensure reproducibility.

### 3.1 DATASETS

**CLOTHO** (Drossos et al., 2020) comprises audio recordings with durations ranging from 15 to 30 seconds, each accompanied by captions containing 8 to 20 words. The development set is divided

Table 3: Combination coefficients for four systems

| SID | 2 | | | 3 | | |
|---|---|---|---|---|---|---|
| Model | PaSST | EAT | BEATs | PaSST | EAT | BEATs |
| E1 | 0.2275 | 0.07 | 0.06 | 0 | 0.12 | 0.045 |
| E2 | 0.2275 | 0.0875 | 0.04 | 0 | 0.15 | 0.03 |
| E3 | 0.225 | 0.175 | 0.1 | 0.03 | 0.01 | 0.01 |
| E4 | 0.18 | 0.14 | 0.08 | 0.09 | 0.03 | 0.03 |
| SID | 4 | | | 5 | | |
| Model | PaSST | EAT | BEATs | PaSST | EAT | BEATs |
| E1 | 0.325 | 0 | 0.045 | 0.0975 | 0.01 | 0 |
| E2 | 0.325 | 0 | 0.03 | 0.0975 | 0.0125 | 0 |
| E3 | 0.195 | 0.045 | 0.06 | 0.09 | 0.03 | 0.03 |
| E4 | 0.13 | 0.03 | 0.04 | 0.15 | 0.05 | 0.05 |

into training, validation, and test splits. Each recording is paired with five captions created by human annotators.

**AudioCaps** (Kim et al., 2019) consists of 51,308 audio recordings sourced from AudioSet, each 10 seconds long and paired with a single human-generated caption. The captions have an average length of 9.8 words. For our experiments, we combined the training, validation, and test splits of AudioCaps into a single dataset, which was used for pretraining the model.

**WavCaps** (Mei et al., 2024) is a weakly-labeled dataset containing 403,050 audio recordings of varying durations, collected from sources including FreeSound, BBC Sound Effects, SoundBible, and the strongly supervised subset of AudioSet. To adhere to this year's updated competition rules, we excluded any recordings in WavCaps that overlapped with the evaluation subsets of Clotho and were used for pretraining as well.

## 3.2 AUDIO EMBEDDING MODELS

**The Patchout faSt Spectrogram Transformer (PaSST)** (Koutini et al., 2021) leverages pre-trained parameters from a vision transformer and fine-tunes them on the AudioSet dataset for general-purpose audio tagging. By dropping patches from the input sequence, PaSST achieves a low computational and memory footprint. In our experiments, we used a PaSST version without patch overlap, applying structured patchout of 2 and 15 over the frequency and time dimensions, respectively.

**The Efficient Audio Transformer (EAT)** (Chen et al., 2024) is an audio self-supervised learning (SSL) model focused on efficient representation learning from unlabeled audio data. It employs a novel Utterance-Frame Objective (UFO) that combines global utterance-level and local frame-level learning to improve audio understanding. We initialized the models with publicly available pretrained weights, namely EAT-base_epoch30_pt.

**Bidirectional Encoder representation from Audio Transformers (BEATs)** (Chen et al., 2022) is a self-supervised learning framework designed for pre-training comprehensive audio representations. It integrates an acoustic tokenizer with an audio SSL model, optimized iteratively to generate discrete labels rich in audio semantics. We also initialized BEATs with publicly available pretrained weights, namely BEATs_iter3_plus_AS2M.

## 3.3 SENTENCE EMBEDDING MODELS

**RoBERTa** (Liu et al., 2019) is a BERT-based language model developed by Facebook AI that improves upon the original BERT pre-training methodology. By removing the Next Sentence Prediction (NSP) objective, extending training duration, increasing batch size, and leveraging a larger and more diverse corpus, RoBERTa achieves stronger performance in sentence-level representation learning. In our experiments, we used RoBERTa-large as a sentence embedding extractor, utilizing its pretrained parameters to capture rich semantic information from textual inputs.

## 3.4 TRAINING

We preprocess audio to match each model's pretraining setup, train with AdamW, and apply a cosine-warmup scheduler. Specifically, EAT and BEATs used a sampling rate of 16 kHz, while PaSST used 32 kHz. In all cases, audio was converted to log-mel spectrograms as the input representation. All models were trained using the AdamW optimizer. Learning rates were adjusted using a cosine warmup scheduler, with specific values detailed in the respective training stages. The training process was divided into three stages. Initial pretraining was conducted on the CLOTHO, WavCaps, and AudioCaps datasets to learn general audio-text alignment, while the subsequent finetuning and re-finetuning stages were performed exclusively on the CLOTHO dataset. Each stage is described below.

**Initial pretraining** – We use a mix of CLOTHO development training split, AudioCaps, and WavCaps datasets. The training spans 20 epochs. No data augmentation is applied in this phase. Due to computational resource constraints, we set batch size to 64 for PaSST, 24 for EAT, and 16 for BEATs. To accommodate these configurations, we adjusted the learning rates using a cosine warmup scheduler across all training processes. For PaSST, the learning rate decreased from 2e-5 to 1e-7, while for EAT and BEATs, it decreased from 1e-5 to 1e-7. These hyperparameter settings were consistently applied in the subsequent finetuning and re-finetuning stages.

**Finetuning** – In the finetuning phase, models were further trained for 20 epochs using ensemble soft labels. We compute soft labels by averaging similarities from three audio models (Eq. 5) and train with a distillation loss. To enhance robustness, we also apply back-translation, LLM-based caption mixing, and one-word random deletion or synonym replacement with 0.8 probability.

**Re-finetuning with cluster-guided classification** – In the re-finetuning phase, we enhanced our model through cluster-guided classification. We perform clustering using two sets of weights: our finetuned model and e5-large-v2 weights, sourced from the e5 model family and utilized within the BERTopic framework [5, 18]. The e5-large-v2 model excels in clustering tasks by generating high-quality sentence embeddings that preserve semantic similarity in the embedding space. For each embedding set, we employed the BERTopic framework with HDBSCAN to assign pseudo-labels to text samples, reassigning outliers based on topic probabilities estimated by BERTopic. Re-finetuning spanned 20 epochs.

We evaluated four systems combining pretraining, distillation, caption augmentation, and cluster supervision. The configuration of these variants is summarized in Table 1.

## 4 RESULTS

Table 2 presents the performance of our four systems on the CLOTHO development test split. The systems, detailed in Table 1, vary in their use of distillation, data augmentation, and clustering, with three audio models. PaSST consistently outperformed EAT and BEATs across all systems, achieving the highest mAP@16.

A weighted ensemble of Systems 2–5 substantially improved performance over individual systems. We consider two strategies: (E1-E2) system-level then model-level weighting; (E3-E4) the reverse. We select weights via grid search on the validation set. By leveraging the complementary strengths of the systems and models, the ensembles achieved a highest mAP@16 of 48.83.

For the final evaluation, we retrained all systems on the entire development split of the CLOTHO dataset and computed the weighted sum of their similarity matrices using the weights from Table 3. This approach achieved mAP@16 of 0.421 on the evaluation dataset.

## 5 CONCLUSION

This paper presents a novel system for text-grounded audio retrieval. Drawing inspiration from state-of-the-art methodologies, we applied data augmentation techniques leveraging LLMs and incorporated a distillation loss to enhance model performance. Furthermore, by utilizing clustering, we introduced an auxiliary classification task to the training process, which contributed to additional performance gains. These strategies improved retrieval performance.

Limitations include reliance on proprietary LLMs for augmentation and mixed single-model gains from cluster supervision; future work will replace closed components and study memory-augmented contrastive learning under small-batch regimes.

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

## A   THE USE OF LARGE LANGUAGE MODELS

In the preparation of this manuscript, we utilized a large language model (LLM) to assist with refining and polishing the language of our text. Specifically, the LLM was employed to enhance the clarity, conciseness, and readability of sentences, ensuring that our ideas were communicated effectively. No content generation, idea formulation, or substantial writing was performed by the LLM; its role was strictly limited to stylistic improvements under human supervision.

