# OpenReview forum: "LLM-Augmented Soft-Label Distillation and Cluster-Guided Alignment for Language-Based Audio Retrieval"
_ICLR.cc/2026/Conference — ICLR 2026 Conference Withdrawn Submission_

### Official Review · Reviewer_aQdA · 2025-10-27

**Soundness:** 2
**Presentation:** 2
**Contribution:** 1
**Rating:** 0
**Confidence:** 5

**Summary:**

The paper combines several known engineering tricks to improve audio retrieval performance. The level of innovation and paper writing feels more like a workshop paper rather than a proper ICLR paper candidate.

**Strengths:**

The author acknowledges the use of LLM during paper writing.

**Weaknesses:**

The paper is missing many important details. For example, what do E1, E2, E3, and E4 mean in Table 3? And what does "system-level" weighting mean?
The paper needs significant improvement on both writing and experimental design.

**Questions:**

N/A

---

### Official Review · Reviewer_tZbx · 2025-10-27

**Soundness:** 1
**Presentation:** 1
**Contribution:** 1
**Rating:** 0
**Confidence:** 5

**Summary:**

This paper proposes a text-to-audio retrieval model with three improvements: 1) soft label distillation from an ensemble of retrieval teachers, 2) LLM-based caption augmentation, including back translation and caption mix, and 3) cluster-guided auxiliary classification.

**Strengths:**

This paper shows a model with three improvements and performs well on Clotho dataset.

**Weaknesses:**

This submission might fit a DCASE challenge track. As presented, it does not meet ICLR standards for novelty, comparison to prior work, and clarity of experimental protocol.

1. Missing problem background. The introduction jumps quickly to a recipe without framing the research questions or the nature of audio text correspondence ambiguity beyond brief statements.
2. No Related Work section. Although citations appear, there is no dedicated related work discussion that situates soft label distillation, clustering for retrieval, or LLM based augmentation relative to prior audio retrieval and cross modal alignment literature.
3. Lack of external baselines and comparisons. Results are only internal variants and ensembles on Clotho. There is no quantitative comparison to prior published systems on the same split or to widely used retrieval models, which makes it hard to judge novelty or impact.
4. Ambiguous reporting of evaluation settings. Table 2 mixes “Multiple annotation” and “Single annotation” metrics, but the paper does not define these terms or provide the exact protocol for producing two mAP@10 values. The table header shows two mAP columns and the text around it offers no definitions.
5. Confusing experiment protocol. In the final paragraph of the results section. It is confusing that the paper is stating: "we retrained all systems on the entire development split of the CLOTHO dataset and computed the weighted sum of their similarity matrices using the weights from Table 3." (What is entire development split? Is it development train split or test split or both?) "This approach achieved mAP@16 of 0.421 on the evaluation dataset." (Before this sentence the mAP@16 reported is multiplied by 100, e.g. "the ensembles achieved a highest mAP@16 of 48.83", why here it has a different scale?)

**Questions:**

See above

---

### Official Review · Reviewer_K79C · 2025-11-01

**Soundness:** 1
**Presentation:** 1
**Contribution:** 1
**Rating:** 0
**Confidence:** 3

**Summary:**

This paper proposes a combination of techniques namely soft-label distillation, LLM-based data augmentation, and cluster-guided alignment for language-based audio retrieval. While the core problem of handling ambiguous audio-text correspondences is significant and the individual methods are reasonable, the paper suffers from critical flaws that severely undermine its contribution. Another issue is that the authors' own ablation study appears to invalidate their core contributions: the proposed LLM augmentation and cluster-guided methods show no consistent improvement over a simpler baseline that uses only distillation. Combined with the complete lack of comparison to existing state-of-the-art methods and a manuscript that reads like an unfinished draft, these issues prevent the paper from making a convincing case for a meaningful contribution.

**Strengths:**

The focus on handling ambiguous, non-binary audio-text correspondences is a relevant and valuable direction for the field. The techniques employed, such as using LLMs for semantically-aware caption mixing and employing clustering for auxiliary supervision, are conceptually sound and interesting ideas worthy of exploration.

**Weaknesses:**

The paper is structurally incomplete, lacking a proper introduction, related work section, and sufficient detail to be considered a finished submission. This makes a thorough evaluation difficult.  The results in Table 2 indicate that all meaningful gains are attributable to the distillation technique (SID 2). The subsequent addition of LLM augmentation (SID 3) shows a performance drop for the best-performing PaSST model, and the cluster-guided alignment (SID 4/5) fails to recover this loss or provide a clear improvement. This directly challenges the utility of the two techniques highlighted in the title and abstract.

**Questions:**

The manuscript is only 6 pages and lacks a related work section and a substantial introduction. Can you confirm this is the final version intended for review? How do you justify these structural omissions?

---

### Official Review · Reviewer_ksuK · 2025-11-04

**Soundness:** 2
**Presentation:** 1
**Contribution:** 2
**Rating:** 0
**Confidence:** 5

**Summary:**

This paper proposes an enhanced dual-encoder framework for text-audio retrieval task with two techniques: Soft-label distillation and Cluster-guided Alignment with auxiliary classification. It leverages an ensemble of teacher models to produce non-binary similarity distributions that capture ambiguous correspondences between audio and captions. Then it applies GPT-4o for back-translation and “LLM-mix” caption generation to expand the caption diversity. Finally it conducts a BERTopic-style clustering on captions to assign pseudo-labels, which are then used as an auxiliary classification objective for both audio and text encoders. The approach is evaluated on the Clotho dataset and the ablation tables show that distillation yields the largest gain. And clustering method gives mixed results depending on backbones (i.e, BEAT, PASST, EAT).

**Strengths:**

The strength of this paper lies in the motivation. It has clear motivation on improving the text-audio results by enhancing the model's capability in understanding the gap from the predictions to the ground-truths. The paper recognizes that binary audio–caption pairs fail to capture nuanced correspondences (e.g., overlapping acoustic events). The idea of using ensemble soft targets for distillation is reasonable and aligns with prior retrieval improvements in [1].

[1] Estimated Audio-Caption Correspondences Improve Language-Based Audio Retrieval, DCASE technical report, 2024.

**Weaknesses:**

However, the paper has many critical points that hinder its acceptances.

First, the paper has limited novelty. The paper mainly combines existing techniques, soft-label distillation, LLM augmentation, and clustering-based auxiliary tasks, which are more or less proposed in many text-audio tasks (audio captioning, text-audio retrieval, and text-to-audio generations) without any new theoretical formulation or model-wise contribution. Each module closely follows prior work [1][2][3] with light adaptations.

Second, the experimental design is unprofessional. The benchmark comparison is too weak. Only the Clotho dataset is used for evaluation, and it is confusing that the test set of AudioCaps is included in training. The basic benchmark should include both Clotho and AudioCaps test sets for evaluations, the same as many previous text-audio retrieval and audio captioning papers. The text-to-audio retrieval and audio-to-text retrieval performances should split. Nonetheless, the paper does not compare against standard state-of-the-art retrieval or text-audio pretraining frameworks such as CLAP, LAION-CLAP, Comp-A, FLAM, and Audio Flamingo. It has only inner-comparisons among different audio encoders, making it unclear whether the reported numbers represent an advance to other models.

Third, the organization of this paper needs improvement. Figure 1 is difficult to interpret and fails to illustrate the data flow between distillation and clustering modules. The Introduction and Related Work sections are too brief, and the technical sections mix multiple concepts without conceptual clarity. The manuscript reads more like a system-description report than a scientific study, lacking analytical insight and coherent narrative flow.

Last but not least, the theoretical insights of distillation process requires more justification. The proposed soft-label distillation relies entirely on ensemble-generated pseudo-targets and does not incorporate any mechanism to regularize toward ground-truth correspondences. As a result, the convergence behavior of the student model is poorly controlled. It can overfit to the ensemble’s bias without guaranteed correctness. This paper feels more like an empirical “performance trick” in Kaggle (or say DCASE competition ins the audio domain) where you could boost the performance in the last chance. It is rather than a theoretically grounded or generally reliable learning process. The paper does not provide evidence that such soft-target transfer remains effective across datasets or semantic domains, as it is only evaluated on Clotho dataset, where the extreme overfitting might happen.

[1] Estimated Audio-Caption Correspondences Improve Language-Based Audio Retrieval, DCASE technical report, 2024

[2] Improving Audio Captioning Models with Fine-grained Audio Features, Text Embedding Supervision, and LLM Mix-up Augmentation, ICASSP 2024

[3] BERTopic: Neural topic modeling with a class-based TF-IDF procedure, Arxiv Preprint

**Questions:**

1. Why was AudioCaps excluded from evaluation despite being part of the pretraining data?

2. How much improvement does each component (distillation, LLM-augmentation, clustering) yield individually on the same backbone? Some combinations of the ablation studies are presented but the individual effectiveness of these three components (or say two components) are not presented.

3. How does the method compare quantitatively with CLAP, Comp-A, and Audio Flamingo under identical Clotho retrieval settings (with and without the same training sets)?

4. What are the computational and data costs introduced by GPT-4o augmentation and large-scale clustering?

---

### Note · Authors · 2025-11-19

I have read and agree with the venue's withdrawal policy on behalf of myself and my co-authors.